# Neural Stem Cells Transplanted into Rhesus Monkey Cortical Traumatic Brain Injury Can Survive and Differentiate into Neurons

**DOI:** 10.3390/ijms25031642

**Published:** 2024-01-29

**Authors:** Shuyi Liu, Liping Shi, Tianzhuang Huang, Yuyi Luo, Yongchang Chen, Shangang Li, Zhengbo Wang

**Affiliations:** 1State Key Laboratory of Primate Biomedical Research, Institute of Primate Translational Medicine, Kunming University of Science and Technology, Kunming 650500, China; lpbrliushuyi@163.com (S.L.); sleeping0527@163.com (L.S.); huangtz@lpbr.cn (T.H.); luoyuyi2021@163.com (Y.L.); lis101@163.com (S.L.); 2Yunnan Key Laboratory of Primate Biomedical Research, Kunming 650500, China

**Keywords:** superficial cerebral cortex, synaptic connections, long-term beneficial, glial scar formation

## Abstract

Cortical traumatic brain injury (TBI) is a major cause of cognitive impairment accompanied by motor and behavioral deficits, and there is no effective treatment strategy in the clinic. Cell transplantation is a promising therapeutic strategy, and it is necessary to verify the survival and differentiation of cells after transplantation in large animal models like rhesus monkeys. In this study, we transplanted neural stem cells (NSCs) and simultaneously injected basic fibroblast growth factor/epidermal growth factor (bFGF/EGF) into the cortex (visual and sensory cortices) of rhesus monkeys with superficial TBI. The results showed that the transplanted NSCs did not enter the cerebrospinal fluid (CSF) and were confined to the transplantation site for at least one year. The transplanted NSCs differentiated into mature neurons that formed synaptic connections with host neurons, but glial scar formation between the graft and the host tissue did not occur. This study is the first to explore the repairing effect of transplanting NSCs into the superficial cerebral cortex of rhesus monkeys after TBI, and the results show the ability of NSCs to survive long-term and differentiate into neurons, demonstrating the potential of NSC transplantation for cortical TBI.

## 1. Introduction

Traumatic brain injury (TBI) is mainly caused by an external mechanical force [1] that induces neuronal death at the site of injury, resulting in diffuse axonal injury [2]. TBI can lead to motor and cognitive impairment and even to long-term physical disability [3]. The main treatments for TBI include hyperbaric oxygen therapy, non-invasive brain stimulation, task-oriented functional electrical stimulation, and physiotherapy [4,5]. However, none of these treatments can effectively reverse neurological deficits [6]. Therefore, searching for a treatment strategy that can replace the dead brain tissue is necessary.

Several studies assessing the effectiveness of stem cells in treating TBI have been conducted [7,8]. Neural stem cells (NSCs) have a self-renewal ability and can differentiate into mature neurons [9,10]. Studies on the effectiveness of NSCs in treating TBI in rodents have shown that transplanted NSCs can survive and differentiate into neurons and then partially replace the damaged neurons, resulting in improved motor function after TBI, that a study in TBI rat models showed that NSCs are capable of surviving engraftment and differentiating into neurons which, in turn, correlate with improvements in neurological recovery [11,12,13,14]. Additionally, a study in mice also showed that the transplantation of NSCs can potentially repair and integrate neurons and glial cells at the injury site [15]. These results suggest that NSC transplantation has potential as a clinical treatment of TBI. Our previous study showed that transplanted NSCs could survive and differentiate into functional neurons in the inferior colliculus, a deep brain region, in injured rhesus monkeys [16]. Further, bFGF/EGF was found to improve the survival and differentiation of transplanted cells in the rat brain in our previous study [17]. At the same time, apart from cell replacement and transplant-mediated deficit amelioration, transplant location-dependent neuroprotection may be key to delaying the onset or preventing the development of injury-induced disability in rats [18]. Therefore, we transplanted NSCs into the cortex in this study based on concepts that were established in previous research. In filling cortical damage areas via transplantation of NSCs, it is essential that the transplanted cells are not removed via the CSF and that they survive and differentiate into neurons at the site of injury. In this study, a model of superficial cortical injury was established, and NSC transplantation combined with bFGF/EGF injection was performed at the damaged site. The NSCs were observed one-year post-transplantation to determine whether they could survive in situ and differentiate into neurons. This study may provide an idea for future research on brain injury.

## 2. Results

### 2.1. Induction and Identification of NSCs

NSCs are derived from embryonic stem cells (ESCs) of rhesus monkeys; the details of the induction process are presented in the Section 4. The NSCs were positive for NSC marker (Nestin, sex-determining region Y-box 2 (SOX2)) (Appendix A) and negative for neural markers (microtubule-associated protein 2 (MAP2) and neuronal Class III β-tubulin (Tuj-1)) (Appendix A). In addition, Western blot analysis is also used to identify NSCs (Appendix A). The differentiation capacity of NSCs was assessed by staining for neural markers (MAP2 and Tuj-1), including markers for dopaminergic (DA) neurons (tyrosine hydroxylase (TH) and Girk2) (Appendix A).

### 2.2. Survival and Differentiation of Grafted NSCs in the Site of TBI One Year after Transplantation

A rhesus monkey TBI model was created by injuring the visual and sensory cortices using the technique described in a previous study [16]. Nestin^+^/(sex-determining region Y-box 2 (SOX 2^+^)) NSCs combined with bFGF/EGF were delivered to the injury site seven days after brain injury (Figure 1A). The site of cortical damage before transplantation appeared as a hole in Figure 1Ba.

Transplanted NSCs stayed in the injury site for one year and showed a “U”-shaped structure in the macaques (Figure 1Bb). Immunohistochemical studies were performed after cell transplantation, and the results showed that the transplanted NSCs could survive for at least one year (Figure 1C–E). The neuronal identity of the surviving transplanted GFP^+^ cells was then confirmed by staining for the classic neural markers Tuj-1, neurofilament (NF), and neuronal nuclei (NeuN). Figure 1C,D shows GFP^+^ cells fused with Tuj-1^+^ and NF^+^ nerve fibers and that the mature neuron marker NeuN was also expressed in the nuclei of the GFP^+^ cells. The results showed that the transplanted NSCs could differentiate into neurons (Figure 1E).

The same experiment was performed in the sensory cortex. The density of GFP^+^ cells was similar to that in the visual cortex (Figure 2A), and the GFP^+^ cells expressed Tuj-1. The GFP^+^ cells also expressed NF (Figure 2B), suggesting that some NSCs had differentiated neurons that formed into mature nerve fibers. NeuN was also expressed in the nuclei of the GFP^+^ cells, as in the visual cortex (Figure 2A–C). Statistical analysis showed that there was no significant difference in the expression of NeuN or NF between the visual cortex and the sensory cortex (Figure 2D,E). These results demonstrated that the transplanted GFP^+^ NSCs could survive for at least one year and differentiate into neurons in the sensory cortex and the visual cortex.

### 2.3. Activity and Migration of the Grafted NSCs

Although the majority of transplanted cells in the visual cortex and sensory cortex remained at the transplant sites (damage sites), some of the transplanted cells migrated away from the damage sites as individual cells or clusters at the border between the host brain tissue and the graft. c-Fos is an immediate early gene and proto-oncogene that is highly expressed in active cells and is useful for studying cellular activity in the brain. c-Fos is co-expressed with neuronal markers (NF, NeuN), and staining for c-Fos is used to determine whether neurons differentiated from transplanted NSCs are active [19]. Figure 3A shows that c-Fos was expressed in both GFP+ and GFP^−^ cells (host cells), and as shown in Figure 3B,C, NF/Tuj-1-positive nerve fibers comprised GFP+ and GFP^−^ cells. The staining patterns of c-Fos, NF, and Tuj-1 did not differ between host neurons and GFP^+^ neurons. 

Synapsin I, a major phosphoprotein found in synaptic vesicles and located in the presynaptic membrane, was used as a marker to identify presynaptic terminals [20]. Some presynaptic terminals of host neurons (Synapsin I^+^/GFP^−^) were identified within the graft (Figure 4A). This result, taken together with the postsynaptic density protein-95 (PSD-95) [21] staining data, showed that a large number of host-neuron-derived synapses were formed on differentiated neurons in the transplantation site (Figure 4C), and the enlarged figures of the edge of grafts also show potential connections to host tissue (Figure 4B,D). These results possibly provide an anatomical basis for integration between host neurons and transplanted NSCs.

The results of immunohistochemical staining in the sensory cortex were similar results to those of immunohistochemical staining in the visual cortex. The transplanted GFP^+^ NSCs expressed c-Fos, Tuj-1, and NF (Figure 5A–C). There was no clear border at the graft edge. The GFP^+^ fibers were partially fused with the host tissue. Statistical analysis showed that there was no significant difference in the expression of c-Fos between the visual cortex and the sensory cortex (Figure 5D). These results indicated that transplanted NSCs could differentiate into neurons with cellular activity in both the visual cortex and sensory cortex. The details of antibodies showed in Appendix A. 

Overall, these immunohistochemical data showed that NSCs survived after transplantation, that the cells differentiated into mature neurons, and that there was potential integration between differentiated and host neurons.

### 2.4. Differentiation of Grafted NSCs into Astrocytes

In vitro, NSCs can differentiate into glial cells, which express markers of astrocytes (glial fibrillary acidic protein (GFAP)). Therefore, GFAP staining was also performed on graft tissue sections. Immunohistology showed that in both the visual cortex and sensory cortex, some GFP^+^ cells expressed GFAP, indicating that some of the grafted NSCs differentiated into glial cells. However, no glial scar was observed at the edge of the injury site (Figure 6A,B), and statistical analysis showed that there was no significant difference in the expression of GFAP between the visual cortex and the sensory cortex, as shown in Figure 6C. In addition, the expression of GFAP at the edge of the grafts both in the visual cortex and the sensory cortex showed no obvious glial scars in both GFP^+^ and GFP^−^ tissues (Figure 7A,B). These results suggested that the formation of connections between newborn neurons and host neurons was not impeded and demonstrated the feasibility of NSCs to repair the damage sites.

### 2.5. Differentiation of Grafted NSCs into Dopaminergic Neurons

Our in vitro experiment and previous animal studies have shown that transplanted NSCs can differentiate into specific neurons, including DA neurons (Figure 2A,B). Therefore, the distribution of DA fibers in the visual cortex was analyzed by staining for TH. The results showed that GFP^+^ cells partially differentiated into DA neurons, and fibers extending from host tissue could be observed in the graft (Figure 8). There was no clear boundary between GFP^+^/TH^+^ cells and GFP^−^/TH^+^ cells. This result indicated widespread differentiation of transplanted NSCs not only into mature neurons but also into special types of neurons. The details of above antibodies used in results showed in Appendix A. 

## 3. Discussion

TBI is a global health issue for which effective therapies are lacking. In recent years, extensive research on the effectiveness of NSCs in the treatment of TBI has been conducted. Cortical TBI usually results in a larger lesion cavity, and transplanted cells can easily enter the CSF. Therefore, a previously developed mechanical injury method [16] was used to damage the visual and sensory cortices, and then NSCs were transplanted into the injury site. The results showed that the transplanted NSCs did not enter the CSF, were confined to the transplanted site, filled in the injury site, and survived for at least one year. The surviving engrafted cells differentiated into mature neurons, including DA neurons. There was no clear boundary between the periphery of the graft and the host tissue, and there was no glial scar at the border. Furthermore, the transplanted NSCs differentiated into mature neurons with cellular activity and formed synaptic connections with host neurons.

In a previous study, transplantation of human fetal neural progenitor cells (hfNPCs) into lesions was found to reduce reactive astrogliosis, but only a few of the transplanted hfNPCs differentiated into neurons [22]. Similarly, Haus et al. transplanted hNSCs into the lesion in an immunodeficient athymic nude rat model of controlled cortical impact and found that the transplanted hNSCs could differentiate into neurons, astrocytes, and oligodendrocytes [12]. Regarding survival and differentiation after transplantation, neurotrophic factors such as brain-derived neurotrophic factor (BDNF) and B-cell lymphoma-extra-large (Bcl-xL) have been shown to play a key role in differentiation after transplantation [12,23,24]. Our previous studies in rats have also shown that the administration of bFGF/EGF in combination with transplanted NSCs improves cell survival and the rate of neuronal differentiation [17]. In this study, NSCs transplanted into the injured cortex (including the visual cortex and sensory cortices) survived for at least one year and differentiated into mature neurons. There was no significant difference in neuronal or glial cell differentiation between the visual cortex and the sensory cortex. Unlike in previous studies, NSCs transplanted in rhesus monkeys survived and differentiated into neurons in large numbers in this study. These findings in rhesus monkeys demonstrate the advantages of NSC transplantation combined with bFGF/EGF injection in brain injury [25]. In addition, this is the first time that NSCs were transplanted into the superficial cortex of macro-animal rhesus monkeys with TBI, and the results showed that NSCs were confined to the transplantation site and successfully differentiated and matured into active neurons, which proved that NSCs could survive and differentiate well in situ.

Previous studies in rats have shown that the integration of transplanted NSCs into the host neural circuitry is important for improving the efficacy of NSC transplantation [26,27]. Recent studies have also shown that transplanted human brain organoids can structurally and functionally integrate into the visual system after cortical injury, demonstrating the importance of functional integration in the treatment of TBI by cell transplantation [28]. In this study, the transplanted NSCs not only showed cellular activity but also expressed PSD-95 and Synapsin I, indicating that there were potential synaptic connections between transplanted NSCs and host neurons. These results suggest that transplanted NSCs could survive in situ and differentiate into neurons with synaptic connections. Although the injured areas were small and no significant behavioral changes were observed, these results still suggest a long-term beneficial effect of NSC transplantation in the repair effect after TBI.

Previous studies have shown that TBI triggers secondary neuroinflammation, reactive gliosis, and even glial scar formation in situ and that the transplantation of NSCs can reduce the inflammatory response [29,30,31]. In this study, the transplantation was performed seven days after mechanical injury. The results showed that only a small number of GFAP^+^ cells were present one year after transplantation and that there was no glial scar, which suggested the safety of NSC transplantation.

Previous studies using NSC transplantation for the treatment of TBI have shown the effectiveness of this treatment strategy, but most of these studies were conducted in rodent TBI models. In our previous study, NSCs transplanted into deep brain areas in rhesus monkeys after injury survived and differentiated into neurons, but the effect of transplanting NSCs into superficial brain areas in monkeys has not been explored. This study was the first to transplant NSCs into the superficial cortex of rhesus monkeys after TBI, and the results showed that the transplanted NSCs remained in the lesion site. Moreover, the simultaneous application of bFGF/EGF allowed the transplanted NSCs to survive long-term and differentiate into neurons without inducing glial scar formation, which can hinder recovery after TBI. At present, the clinical potential of NSC transplantation in the treatment of TBI needs to be better understood [32]. This study supports the feasibility and safety of NSC transplantation for cortical injury filling. However, this study also has limitations. First of all, since our study was initially designed to observe the effect of NSC transplantation on the restoration of the six layers of the cortex, but it was not observed in the final results, for this reason, we were not able to evaluate the functional aspects, such as electrophysiology. In addition, although we tested the behavioral phenotypes of rhesus monkeys after both mechanical injury and NSC transplantation, no significant behavioral changes were observed due to the small injury area and the strong compensatory ability of monkeys. Therefore, behavioral detection data were not displayed in the results of this paper. In subsequent studies, we will expand the volume of mechanical damage and further select brain regions that have greater and more obvious effects on behavior so as to contribute to behavioral research.

## 4. Materials and Methods

### 4.1. Embryonic Stem Cell Culture

GFP-labeled rhesus ESCs were incubated in ES medium consisting of Knockout DMEM, 20% KO-SR, 1% nonessential amino acids, 2 mM L-glutamine (Life Technologies, Carlsbad, CA, USA), 0.1 mM β-mercaptoethanol (β-Me), and 10 ng/mL basic fibroblast growth factor (bFGF) (both from Gene Operation, Waltham, MA, USA). ESCs were a gift from the Lyon Stem Cell Research Institute [33]. Lyon-ESCs were co-cultured with mitotically incapable mouse embryonic fibroblasts (CF-1-MEFs; ATCC). CF-1-MEFs were first grown in DMEM containing 2 mM L-glutamine and 15% FBS. The medium was changed daily, and undifferentiated colonies were mechanically passaged every 5–7 days using flame-drawn Pasteur pipettes. All cells were grown at 37 °C in 5% CO_2_.

### 4.2. Induction of NSC Differentiation

Lyon-ESC colonies were first digested with 1 mg/mL neutral protease, washed with ES medium to remove the protease, and then suspended in modified N/M medium (50% DMEM/F12, 50% neural medium, 1× N2 supplement, 1× B27 supplement, and 2 mM L-glutamine) for the induction of NSCs differentiation, then, cells were seeded in 15 mm × 30 mm agar-coated wells (Corning, Corning, NY, USA). The cells were allowed to aggregate for four days to form embryoid bodies (EBs). EBs of uniform size were selected, seeded in extracellular matrix (ECM)-coated four-well plates, and cultured in neuroepithelium (NP) medium for 10–14 days until rosette formation. The NP medium consisted of 50% DMEM/F12, 50% neural medium, 1× N2 supplement, 1× B27 supplement (Life Technologies), 500 ng/mL Jagged-1, 0.2 mM ascorbic acid (AA), 2 ng/mL heparin (PeproTech, Cranbury, NJ, USA) and 2 mM L-glutamine. Rosettes were purified, and the purified NSCs were identified by immunofluorescence.

### 4.3. Induction of Dopaminergic Neurons Differentiation

A two-step differentiation method was used to derive DA neurons from iNSCs. In the first stage [34], NSCs were seeded at a density of 5 × 10^3^ cells per 12 mm glass coverslip that had been coated with ECM in NSC differentiation medium (DMEM/F12, 1× N2 supplement, 1× B27 supplement, 1× NEAA, 2 mM L-glutamine) supplemented with 1 μM SAG1 and 100 ng/mL FGF 8b for 10 days. For stage two, SAG1 and FGF 8b in the medium were replaced with 0.2 mM AA, 0.5 mM cAMP, 10 μM DAPT, 10 ng/mL GDNF, 10 ng/mL BDNF, and 1 ng/mL TGF-βIII (both from Sigma-Aldrich, Darmstadt, Germany) and cultured for another 2 weeks. The differentiated cells were fixed by 4% paraformaldehyde (PFA) on day 24 for analysis.

### 4.4. Immunofluorescence Analysis of NSCs

Rosettes were stained with Nestin, Tuj-1, MAP2, TH, and LMX1A antibodies (all from Abcam, Cambridge, United Kingdom) after purification to check the purity of the NSCs. For immunofluorescence analysis of NSCs, cells were cultured on chamber slides, fixed with 4% PFA for 20 min, and permeabilized with 0.1% Triton X-100 for 15 min. They were then blocked with PBS containing 5% BSA for 1 h at room temperature. The samples were incubated overnight at 4 °C in blocking buffer (1% BSA) containing primary antibody (1:500). The details of antibodies showed in Appendix A and S2. The samples were washed three times with PBS overnight for 5 min each, and the cells were incubated with the corresponding secondary antibody (1:200) for 2 h at 37 °C. The samples were washed with the same wash protocol after secondary antibody incubation and stained with DAPI. Stained samples were examined with an FV1000 fluorescence microscope (OLYMPUS). 

### 4.5. Western Blot Analysis of NSCs

First, the rosette structures were collected and centrifuged at 12,000 rpm/5 min, the supernatant was removed, 200 microliters of Radio-Immunoprecipitation Assay Lysis buffer (RIPA) lysate was added, and protease inhibitor was added at 1:100. The lysate was placed at 4 °C for 40 min, during which the lysed sample was oscillated at 4 °C for 1 min every 10 min. It was then centrifuged at 12,000 rpm for 10 min, after which the supernatant was gently absorbed, transferred to a newly pre-cooled 200 microliter centrifuge tube, and stored at minus 80 degrees. The Bicinchoninic Acid Assay (BCA) kit was used to determine the concentration of protein samples, and the absorbance value at 595 nm was determined by the microplate reader, and the concentration of the protein to be measured was calculated. Then, Western blot analysis was performed according to the concentration of the target protein. First, protein electrophoresis was performed, and the electrophoresis condition was 100 V constant pressure for 90 min. After electrophoresis, the protein was transferred to PVDF under 100 V constant pressure for 30 min. PVDF was placed in 5% bovine serum albumin (BSA) solution after transfer and closed in shakers for 1 h. MAP2, Tuj-1, Sox2, Nestin, and β-actin antibodies were proportionally diluted with 1%BSA (all from Abcam, 1:1000). PVDF was incubated overnight in a 4 °C shaker and incubated the second antibody in a 2 h room temperature shaker the next day. After incubation, TBST was used to clean the secondary antibody, and then PVDF was covered with enhanced chemiluminescence (ECL) for 3 min, and chemiluminescence photos were taken in the gel imager to save the obtained pictures.

### 4.6. Brain Damage Surgery

Five rhesus monkeys (rhesus monkeys, 4–6 years old, weighing 6.5–7.5 kg) were used in this study. All animal procedures complied with the Animal Welfare Act. Our previously developed MRI-guided (3T Siemens Prisma) localization technique with an accuracy of 0.5 mm was used to pinpoint the visual and sensory cortices. The animals were anesthetized using ketamine hydrochloride (10 mg/kg) and sodium pentobarbital (20 mg/kg). Then, the monkey head was fixed to a stereoscope, dental cement was used to fix MRI-visible glass tubes (tubes filled with water) to the skull, and MRI scans were used to locate the visual cortex and sensory cortex.

According to the relative coordinates obtained by MRI, the visual cortex and sensory cortex were mechanically damaged. First, we marked the surface of the skull above the visual cortex and the sensory cortex. A 1.4 mm diameter drill bit is then used to drill through the skull at the mark, creating a catheter-insertion hole. Then, drill four 1 mm reinforcement-screw-embedding holes around the catheter-embedding hole at a radius of 3–4 mm from the center of the catheter-embedding hole (without damaging the dural membrane). The drilled bone hole is flushed with saline to remove the bone residue from the drilled hole. The M1.2 stainless steel screw was inserted into the screw hole, the depth was the thickness of the skull, and the screw was exposed 2–3 mm above the skull. A dural knife was then used to remove the dura inside the catheterization hole, exposing the cerebral cortex. The lower end of the catheter is gently pressed against the cerebral cortex through the catheter holder, and denture cement is poured around the catheter to secure it. After the cement of the denture was solidified, the cerebral cortex was excised (to a depth of about 1.5 mm), and the cut cortical tissue was removed; then, an empty plug was inserted into the catheter. Mechanical injury resulted in a cavity at the injury location. Due to the small volume of the damaged area, we did not find quantifiable behavioral disorders in the observation of TBI monkeys, so this part of the data was not quantified.

### 4.7. Transplantation of NSCs

A week after injury, neural rosettes were digested with trypsin (0.05% in 0.1% EDTA) to form a cell suspension, and the cells were counted. NSCs were suspended in PBS at a concentration of 10^7^ cells per microliter and placed on ice for subsequent cell transplantation. GFP^+^ NSCs were transplanted into the cavities of the visual cortex and the sensory cortex, and the state of each monkey was continuously observed after surgery.

### 4.8. Toluidine Blue Staining

After the mechanical injury, a control monkey was sacrificed, and its brain tissue was collected. Then, the brain was fixed at 4 °C for 4 h and then dehydrated in a sucrose gradient (15%, 20%, and 30%) for cryoprotection. After dehydration, frozen sections were cut at a thickness of 20 µm. Tissue sections were rehydrated and stained with toluidine blue for 20 min, then treated with xylene for transparency and sealed with neutral resin. Images were taken with the Olympus VS200 to observe the tissue structure at the site of mechanical damage.

### 4.9. Identification of Cells by Immunofluorescence

One year after transplantation, four transplanted monkeys were sacrificed, and their brains were collected, fixed at 4 °C for 4 h, and then dehydrated in a sucrose gradient (15%, 20%, and 30%) for cryoprotection. After dehydration, frozen sections were cut at a thickness of 20 µm. The sections were permeabilized with 0.2% Triton X-100 for 40 min and blocked with 5% BSA for 1 h at room temperature. The tissues were permeabilized with 0.2% Triton X-100 for 40 min and blocked with 5% BSA for 1 h at room temperature. The tissue sections were then incubated with primary antibody at 4 °C overnight. The following day, the tissue sections were washed three times and incubated with secondary antibodies for 2 h at 37 °C. The transplanted cells were identified by GFP fluorescence. GFP^+^ cells were then double stained with antibodies against MAP2, NeuN, TUJ-1, NF, TH, c-fos, and glial fibrillary acidic protein (GFAP) (all from Abcam; 1:500) for different purposes, and the transplanted cells were stained with DAPI. Fluorescence was observed using a fluorescence microscope.

### 4.10. Statistics

The NeuN, NF, GFAP, and c-Fos staining data were analyzed using ImageJ. First, open the fluorescent image to be analyzed and separate each channel by Image–Color–Split Channels. Then, select the subsequent measurement area of green fluorescence colocalization pixels by Image–Adjust Threshold and analyze the red and green channel colocalization with the same method, thereby obtaining the colocalization quantitative data. The fluorescent regions of interest for NF, NeuN, and c-Fos were examined in two visual cortex NSC grafts and two sensory cortex NSC grafts, respectively, with two images selected for quantification for each monkey. The proportion of colocalization is expressed by the area of colocalization/green fluorescence area. A *t*-test was performed with GraphPad Prism 5.1 (GraphPad Software, San Diego, CA, USA). Significance was set at *p* < 0.05.

## Figures and Tables

**Figure 1 ijms-25-01642-f001:**
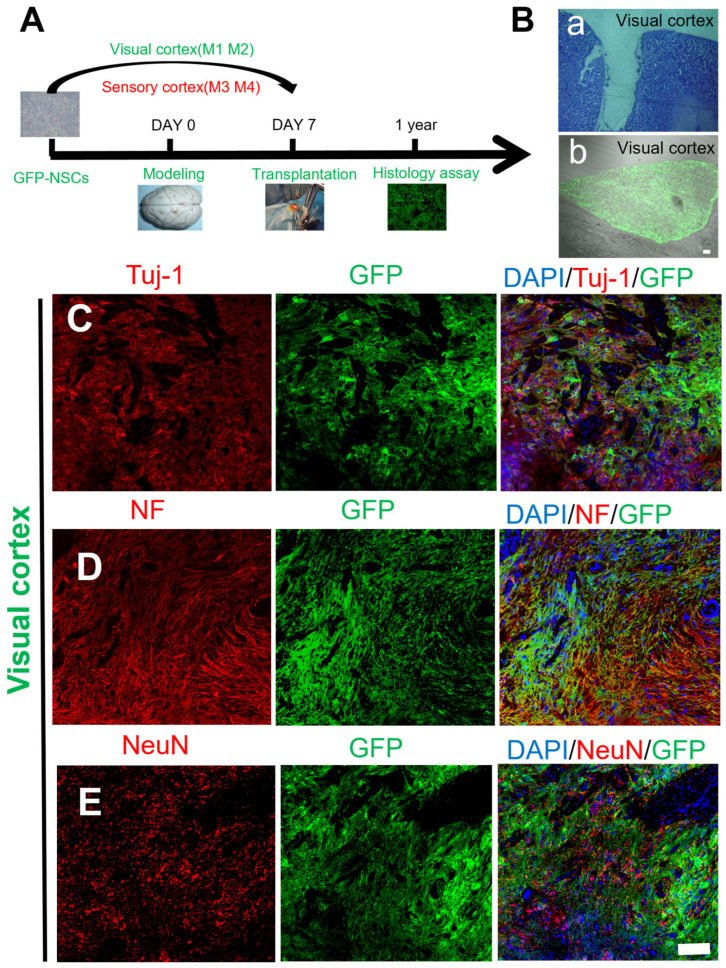
Schematic diagrams showing the mechanical injury protocol and histological images of transplanted GFP-labeled NSCs in the visual cortex. (**A**) Schematic of the mechanical injury protocol and cell transplantation into the macaque sensory/visual cortex. (**B**) a, The lesion site appeared as a hole before transplantation; b, transplantation of GFP-marked NSCs into the mechanical damage of the macaque visual cortex after 1 year. (**C**–**E**) Transplanted NSCs (GFP, green) differentiated into mature neurons in the injured area of the macaque visual cortex (Tuj-1/NF/NeuN, red); scale bar: (**B**): a, b: 100 µm; (**C**–**E**): 100 µm.

**Figure 2 ijms-25-01642-f002:**
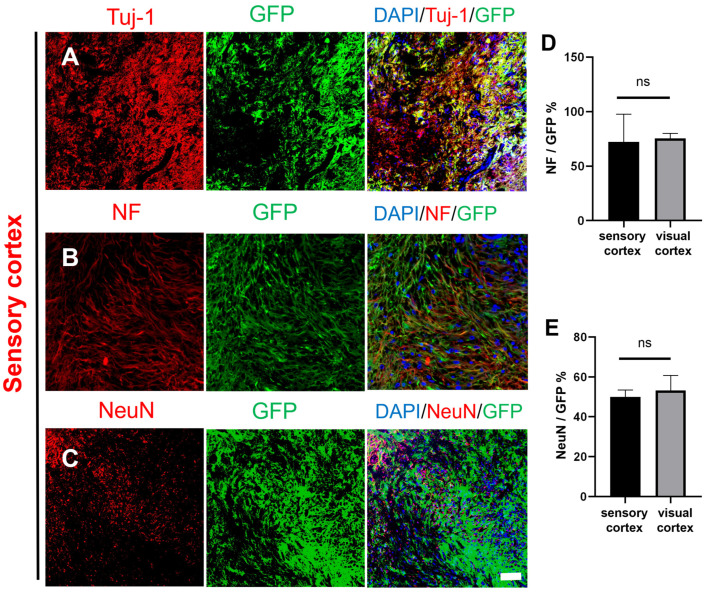
Histological images of transplanted GFP-labeled NSCs in the injured macaque sensory cortex. (**A**–**C**) Transplanted NSCs (GFP, green) differentiated into mature neurons in the injured area of the macaque sensory cortex (Tuj-1/NF/NeuN, red); scale bar, 100 µm. (**D**) NeuN expression was not different between the sensory cortex and visual cortex, and (**E**) NF expression was not different between the sensory cortex and visual cortex; ns denotes *p* > 0.05, means there was no significant difference.

**Figure 3 ijms-25-01642-f003:**
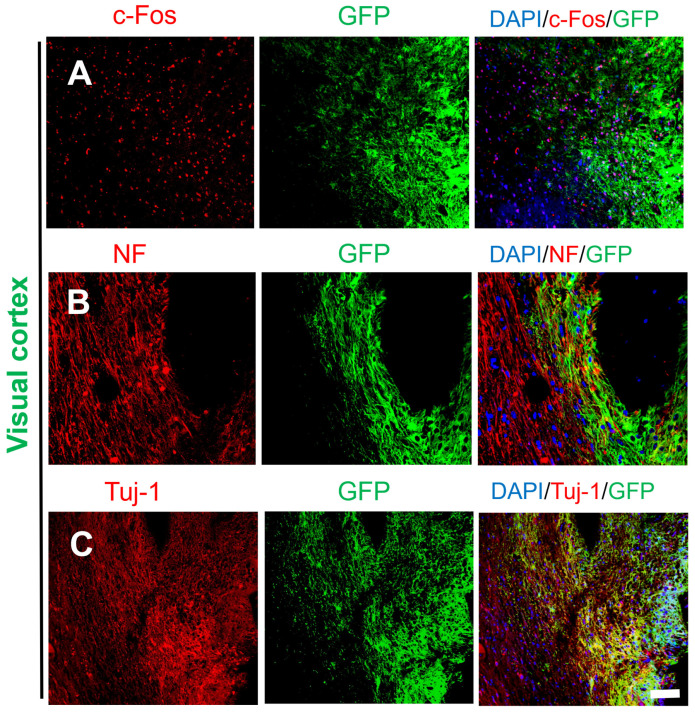
Survival and differentiation of transplanted cells in marginal areas of the damaged visual cortex. (**A**) Some of the transplanted cells were c-Fos+ (DAPI, blue; GFP, green; c-Fos, red). (**B**,**C**) Transplanted NSCs differentiated into mature neurons at the edge of the injury site (DAPI, blue; GFP, green; Tuj-1, red; NF, red); scale bar, 100 µm.

**Figure 4 ijms-25-01642-f004:**
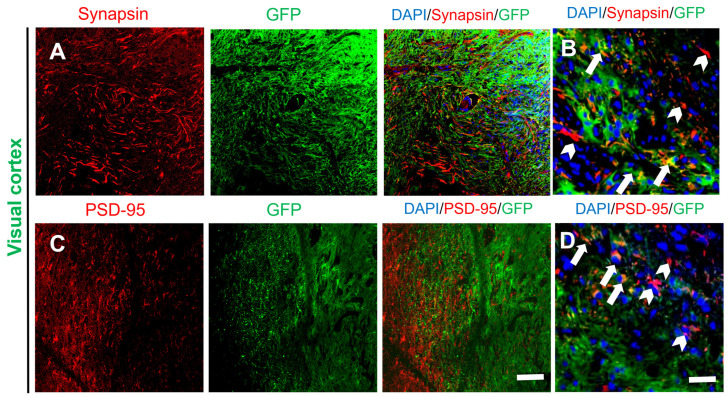
Neurons differentiated from GFP-labeled transplanted NSCs formed synapses with neighboring host neurons. (**A**) Presynaptic terminals of hose and differentiated neurons in the transplantation site (DAPI, blue; GFP, green; Synapsin I, red), and (**B**) the magnified images of the edge between the grafts (arrows) and the host tissue (arrowheads). (**C**) Postsynaptic terminals of differentiated neurons (DAPI, blue; GFP, green; PSD-95, red), and (**D**) the magnified images of the edge between the grafts (arrows) and the host tissue (arrowheads). Scale bars: A, C, 100 µm; B, D, 20 µm.

**Figure 5 ijms-25-01642-f005:**
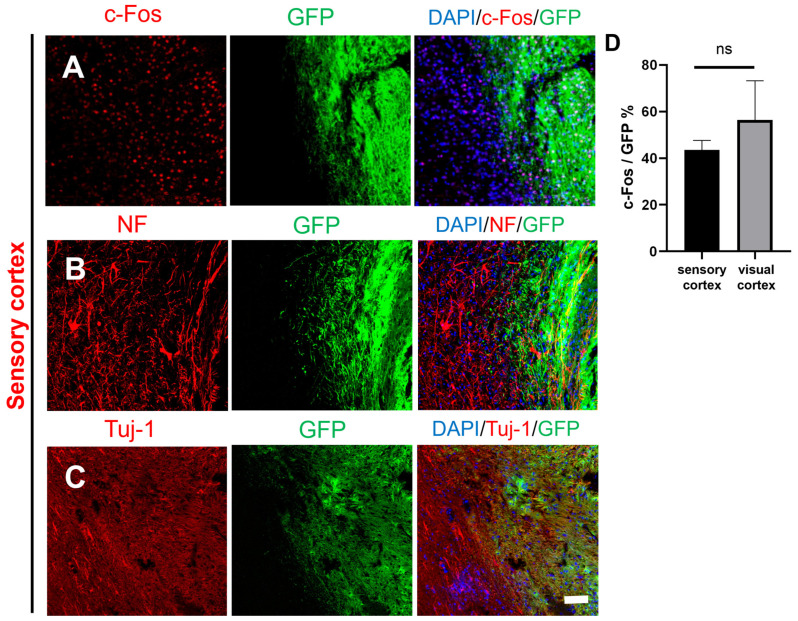
Histological images of transplanted GFP-labeled NSCs in the border between the graft and host tissue in the sensory cortex. (**A**) The transplanted NSCs (GFP) differentiated into active cells (DAPI, blue; GFP, green; c-Fos, red). (**B**–**C**) The transplanted NSCs (GFP) differentiated into mature neurons at the graft edge (DAPI, blue; GFP, green; Tuj-1, red; NF, red); scale bar, 100 µm. (**D**) c-Fos expression was not different between the sensory cortex and visual cortex; ns denotes *p* > 0.05.

**Figure 6 ijms-25-01642-f006:**
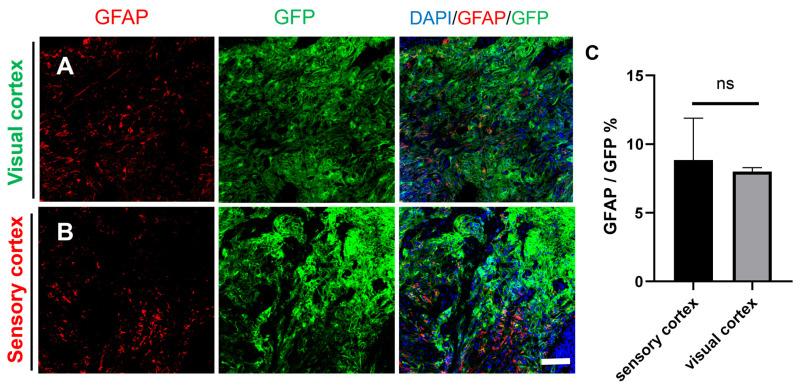
The transplanted NSCs differentiated into astrocytes in the visual cortex and the sensory cortex. (**A**) The transplanted NSCs (GFP) differentiated into astrocytes in the visual cortex (DAPI, blue; GFP, green; GFAP, red). (**B**) The transplanted NSCs (GFP) differentiated into astrocytes in the sensory cortex (DAPI, blue; GFP, green; GFAP, red); scale bar, 100 µm. (**C**) GFAP expression was not different between the sensory cortex and visual cortex; ns denotes *p* > 0.05.

**Figure 7 ijms-25-01642-f007:**
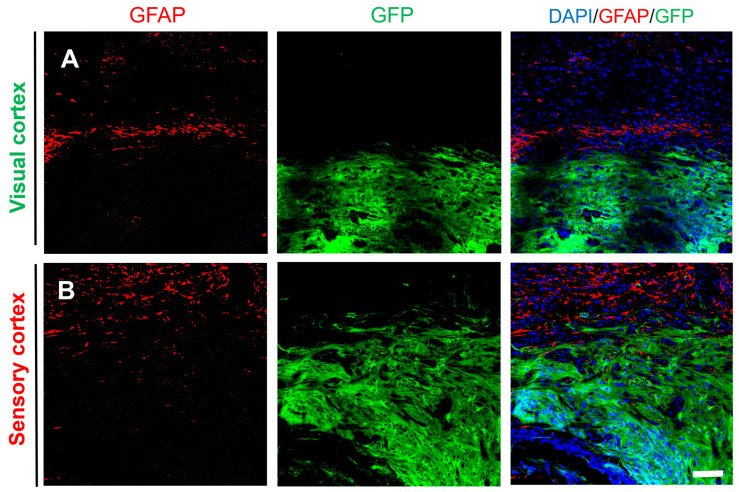
The transplanted NSCs differentiated into astrocytes at the edge of the visual cortex and the sensory cortex. (**A**) The transplanted NSCs (GFP) slightly differentiated into astrocytes both in the grafts and host tissue (DAPI, blue; GFP, green; GFAP, red) and (**B**) in the sensory cortex (DAPI, blue; GFP, green; GFAP, red); scale bar, 100 µm.

**Figure 8 ijms-25-01642-f008:**
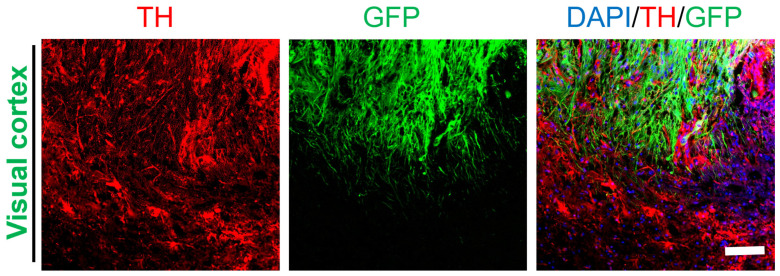
The transplanted NSCs differentiated into DA neurons in the visual cortex. The transplanted NSCs (GFP) differentiated into DA neurons in the visual cortex (DAPI, blue; GFP, green; TH, red); scale bar, 100 µm.

## Data Availability

The data sets used and/or analyzed during this study are available from the corresponding author upon reasonable request.

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
