# Peer review of "Neural Stem Cells Transplanted into Rhesus Monkey Cortical Traumatic Brain Injury Can Survive and Differentiate into Neurons"

_ijms, 2024, doi:10.3390/ijms25031642_

Round 1

Reviewer 1 Report

Comments and Suggestions for Authors

The manuscript presented from Liu et al., entitled ‘Neural stem cells transplanted to the rhesus monkey cortical traumatic brain injury can survive and differentiate into neurons’ is interesting and original, however there are different critical points to be address:

1) Introduction: the authors should improve the text, including more recent references concerning TBI in mammals and other vertebrates.

2) Results: the authors should improve the quality of different figures 1, 2, 3, 5, in particular they should include experimental schemes of the brain regions considered. Concerning neural stem cell marker they should perform further experiments using sox2 to confirm their observations.

I would suggest to verify sox2 expression not only by immunohistochemisty but also by western blot.

3) Material and method: The authors should improve all description concerning the protocol of neural cell differentiation.

4) The conclusions should be supported by a graphical mechanism design.

Reviewer 2 Report

Comments and Suggestions for Authors

ijms-2797803: “Neural stem cells transplanted to the rhesus monkey cortical traumatic brain injury can survive and differentiate into neurons”.

In this study, the authors develop an approach based on a transplantation of neuronal stem cells to treat a traumatic injury of the brain cortex in monkeys. The material is described successively and conclusions are supported by obtained data.

Technical remarks/recommendations:

1)     in lines 16,18,44, the abbreviations should be open;

2)     in lines 45,46 and 50-53, the sentences should be rewritten;

3)     in line 61, “…a previous study” needs a reference;

4)     in all figures, each plate should be denoted by a letter to make easier reference to it in the text;

5)     in lines 142,149,154,159,165,170, “Scale bar, …”;

6)     in lines 144,160,166, “ns denotes p>0.05”;

7)     in line 199, incorrectly positioning reference [27];

8)     in line 216, “…after TBI”;

9)     in line 219, “…the transplantation…”;

10)  in the section “4. Materials and Methods”, the references to the sources for drugs, apparatus, and soft are lacking.

Comments on the Quality of English Language

 Minor editing of English language required

Reviewer 3 Report

Comments and Suggestions for Authors

 The paper deals with the possibility of transplanting neural stem cells to the sites of cortical traumatic brain injuries. Although the results are potentially important in terms of a translational study, there are serious objections to the study design as follows:

1. In the study, 5 rhesus monkeys were used, which were subjected to the same cortical injury. It is not so much a problem that only one animal is a control (it is a very valuable and expensive model) but that it was sacrificed immediately after TBI while the other animals were sacrificed after one year. This is a problem because researchers do not use a control animal anywhere other than to confirm an injury. Thus, they failed to assess the size of the injury in experimental and control animals, as well as the extent of neuroinflammation after one year (say, microglia markers).

2. Although the researchers say that they followed the animals for a year, they did not do any functional or behavioural study to monitor the extent of the injury or possible recovery, so the only thing they evaluate is the survival of the stem cell implants themselves. This is unexpected precisely because it is such an expensive model that functional studies would not be done.

3. The methods are written in insufficient detail to be repeatable. I draw the authors' attention to the fact that they did not state the origin of the stem cells (supplier), there is no manufacturer for any of the chemicals used, especially the antibodies (for which neither the manufacturer nor the concentrations in which they were used are specified), the infliction of mechanical injury is very poorly described, stem cell transplantation not described at all. There is no method description for the images in the supplement. It is not clear from the methods how many animals were used for quantifications, how the images shown were selected and how many animals were quantified or how many NSCs were transplanted per injured place.

4. When displaying the results, the edge between the graft and normal tissue is not visible in pictures 1 and 2, so it cannot be assessed whether it is a graft at all. Images should be displayed at a much lower magnification to show the graft and then at a higher magnification to show the structure of the graft. The authors probably think as they did in Figure 1b, but actually the difference between healthy and transplanted NSCs should be shown at least with DAPI, NeuN and GFAP. On the other hand, the images in the paper, as well as the images in the supplement, were not taken at a high enough magnification to be able to assess whether the authors obtained the usual staining pattern of the investigated epitopes in normal tissue or NCS graft. Also, immunostaining of presynaptic and postsynaptic protein is only proof that these specializations exist, but not that they constitute a functional synapse. It is not clear whether the same animal was used to show implants in the visual and somatosensory cortex, nor whether all 4 animals received implants in both cerebral cortices or whether each animal survived the injury of both regions.

Minor remarks:

1. for a large number of abbreviations, there is no full name or it was not used when the abbreviation was first mentioned.

2. Figure 7 has no scale bar and does not appear to have been taken at the same magnification as the other images.

Round 2

Reviewer 1 Report

Comments and Suggestions for Authors

The authors improved the quality of the manuscript also including new experiments 

Reviewer 3 Report

Comments and Suggestions for Authors

The authors have significantly improved the work, but some key remarks remain as follows:

1. The main objection to the work is still that there is no control - an animal in which a cortical lesion was made, but no NSCs were transplanted after one year. Because of this, an objective assessment of the ability of NSCs to reduce the extent of the lesion or functionally compensate for the loss or influence inflammation at the lesion site is lacking. This mistake can no longer be corrected, but it must be clearly emphasized in the discussion - this means that in the discussion it should be clearly stated what were the expectations, what did not happen (restoration of the 6-layer organization of the cortex) and due to which shortcomings of the study it is not possible to evaluate the reparative effect of NSCs. Also, it should be emphasized in the discussion that proof of functional synapses between the uninjured tissue and the graft must be made electrophysiologically - which is also a limitation of this study. Thus, an entire paragraph of careful breakdown of all the limitations of this study is missing from the discussion.

As a control, you could also use healthy tissue near the graft to show how much the graft (GFP+ cells) differs from the non-graft tissue (GFP-) in terms of marker expression (GFAP/NeuN).

2.  The lack of a behavioral study is another limitation of this research that should be emphasized in the discussion. The authors claim that they have behavioral observations after injury and after NCSs transplantation, but the finding was too small and they state the reasons why it was like that. All this should be stated in the materials and methods, and the explanation for the small finding should be discussed in the discussion chapter, and it should be suggested how a lesion and a behavioral study should be done in the next paper in order to reach measurable results.

3. In your answer under point 3, you state that all 4 animals were used to quantify NSCs in the graft, which gives the impression that you have 4 animals for each measurement. However, under point 4, you state that in 2 out of 4 animals, visible lesions were made, and in 2, lesions of the somatosensory cortex. It follows from this that you only had 2 animals for each measurement shown, so it is not clear how you made statistics from 2 measurements. You could use more than one ROI site per graft for quantification. In the Materials and methods, state clearly how you did the quantification - the number of animals and ROI used for each measurement. Materials and methods still lack a full description of the electrophoresis and Western blot performed on NSCs and shown in the supplement, which refer to procedures with NSCs before transplantation. At the beginning of the results chapter, it would be advisable to separate these results made on NSCs before transplantation from those made on brain tissue after transplantation (which means that 2.1 Results should refer to the content of the supplement).

4.  The justification that you could not show the border between the graft and the uninjured tissue is not valid because the photographs obtained by the microscope can be overlapped and joined. At least in Figure 4 you should show the edge of the graft at a higher magnification to see what you mean when you say that the graft is making synapses with the uninjured tissue ie GFP-negative tissue. Mark the points of interest on the picture.

5. The justification that you could not show the border between the graft and the uninjured tissue is not valid because the photographs obtained by the microscope can be overlapped and joined. At least in Figure 4 you should show the edge of the graft at a higher magnification to see what you mean when you say that the graft is making synapses with the uninjured tissue - GFP-negative tissue. Mark the points of interest on the picture. Pictures should be inserted in appropriate places in individual chapters of the results, and not grouped in chapter 2.5, which should be completely removed from the text as such.

6.  To whom does the following statement mentioned at the end of the work refer:

Informed Consent Statement: Informed consent was obtained from each participant according to 374 the Declaration of Helsinki.

Round 3

Reviewer 3 Report

Comments and Suggestions for Authors

I do not think that this work can be improved more than what is currently achieved. I suggest that it be published in this form.